# Molecularly Targeted Photothermal Ablation of Epidermal Growth Factor Receptor-Expressing Cancer Cells with a Polypyrrole–Iron Oxide–Afatinib Nanocomposite

**DOI:** 10.3390/cancers14205043

**Published:** 2022-10-14

**Authors:** Lekshmi Rethi, Chinmaya Mutalik, Lekha Rethi, Wei-Hung Chiang, Hsin-Lun Lee, Wen-Yu Pan, Tze-Sen Yang, Jeng-Fong Chiou, Yin-Ju Chen, Er-Yuan Chuang, Long-Sheng Lu

**Affiliations:** 1International Ph.D. Program in Biomedical Engineering, Taipei Medical University, Taipei 11031, Taiwan; 2Graduate Institute of Biomedical Materials and Tissue Engineering, Taipei Medical University, Taipei 11031, Taiwan; 3Department of Chemical Engineering, National Taiwan University of Science and Technology, Taipei 10607, Taiwan; 4Department of Radiation Oncology, Taipei Medical University Hospital, Taipei Medical University, Taipei 11031, Taiwan; 5Taipei Cancer Center, Taipei Medical University, Taipei 11031, Taiwan; 6Department of Radiology, School of Medicine, College of Medicine, Taipei Medical University, Taipei 11031, Taiwan; 7School of Medical Laboratory Science and Biotechnology, College of Medical Science and Technology, Taipei Medical University, Taipei 11031, Taiwan; 8Ph.D. Program in Medical Biotechnology, College of Medical Science and Technology, Taipei Medical University, Taipei 11031, Taiwan; 9Graduate Institute of Biomedical Opto Mechatronics, Taipei Medical University, Taipei 11031, Taiwan; 10School of Dental Technology, Taipei Medical University, Taipei 11031, Taiwan; 11Research Center of Biomedical Device, Taipei Medical University, Taipei 11031, Taiwan; 12TMU Research Center of Cancer Translational Medicine, Taipei Medical University, Taipei 11031, Taiwan; 13Department of Medical Research, Taipei Medical University Hospital, Taipei 11031, Taiwan; 14Cell Physiology and Molecular Image Research Center, Taipei Medical University-Wan Fang Hospital, 111, Section 3, Xinglong Road, Wenshan District, Taipei 11696, Taiwan; 15Center for Cell Therapy, Taipei Medical University Hospital, Taipei Medical University, Taipei 11031, Taiwan; 16International Ph.D. Program for Cell Therapy and Regeneration, Taipei Medical University, Taipei 11031, Taiwan; 17TMU Research Center for Digestive Medicine, Taipei Medical University, Taipei 11031, Taiwan

**Keywords:** polypyrrole, iron oxide, afatinib, EGFR, photothermal therapy

## Abstract

**Simple Summary:**

In this manuscript, we describe the design and synthesis of a nanocomposite containing afatinib, polypyrrole, and iron oxide (PIA-NC) to molecularly target epidermal growth factor receptor (EGFR)-overexpressing cancer cells for photothermal conversion. In addition to physical and chemical characterization, we also showed that PIA-NC induces selective reactive oxygen species surge and apoptosis in response to sublethal near-infrared light only in EGFR-overexpressing cancer cells, not in EGFR-negative fibroblasts. The work demonstrates the feasibility of photothermal therapy with cellular precision.

**Abstract:**

Near-infrared–photothermal therapy (NIR-PTT) is a potential modality for cancer treatment. Directing photothermal effects specifically to cancer cells may enhance the therapeutic index for the best treatment outcome. While epithelial growth factor receptor (EGFR) is commonly overexpressed/genetically altered in human malignancy, it remains unknown whether targeting EGFR with tyrosine kinase inhibitor (TKI)-conjugated nanoparticles may direct NIR-PTT to cancers with cellular precision. In the present study, we tested this possibility through the fabrication of a polypyrrole–iron oxide–afatinib nanocomposite (PIA-NC). In the PIA-NC, a biocompatible and photothermally conductive polymer (polypyrrole) was conjugated to a TKI (afatinib) that binds to overexpressed wild-type EGFR without overt cytotoxicity. A Fenton catalyst (iron oxide) was further encapsulated in the NC to drive the intracellular ROS surge upon heat activation. Diverse physical and chemical characterization experiments were conducted. Particle internalization, cytotoxicity, ROS production, and apoptosis in EGFR-positive and -negative cell lines were investigated in the presence and absence of NIR. We found that the PIA-NCs were stable with a size of 243 nm and a zeta potential of +35 mV. These PIA-NCs were readily internalized close to the cell membrane by all types of cells used in the study. The Fourier transform infrared spectra showed 3295 cm^−1^ peaks; substantial O–H stretching was seen, with significant C=C stretching at 1637 cm^−1^; and a modest appearance of C–O–H bending at 1444 cm^−1^ confirmed the chemical conjugation of afatinib but not iron oxide to the NC. At a NIR-PTT energy level that has a minimal cytotoxic effect, PIA-NC significantly sensitizes EGFR-overexpressing A549 lung cancer cells to NIR-PTT-induced cytotoxicity at a rate of 70%, but in EGFR-negative 3T3 fibroblasts the rate was 30%. Within 1 min of NIR-PTT, a surge of intracellular ROS was found in PIA-NC-treated A549 cells. This was followed by early induction of cellular apoptosis for 54 ± 0.081% of A549 cells. The number of viable cells was less than a quarter of a percent. Viability levels of A549 cells that had been treated with NIR or PIA were only 50 ± 0.216% and 80 ± 0.216%, respectively. Only 10 ± 0.816% of NIH3T3 cells had undergone necrosis, meaning that 90 ± 0.124% were alive. Viability levels were 65 ± 0.081% and 81 ± 0.2%, respectively, when only NIR and PIA were used. PIA binding was effective against A549 cells but not against NIH3T3 cells. The outcome revealed that higher levels of NC + NIR exposure caused cancer cells to produce more ROS. In summary, our findings proved that a molecularly targeted NC provides an orchestrated platform for cancer cell-specific delivery of NIR-PTT. The geometric proximity design indicates a novel approach to minimizing the off-target biological effects of NIR-PTT. The potential of PIA-NC to be further developed into real-world application warrants further investigation.

## 1. Introduction

The most prevalent form of lung cancer is non-small-cell lung cancer (NSCLC), at 82% of all diagnoses [1]. Mutations of the epidermal growth factor receptor (EGFR) tyrosine kinase domain lead to constitutively active oncogenic signals and are the most frequent driving mutations for NSCLC [2,3]. Small molecular tyrosine kinase inhibitors (TKI) that bind to mutated EGFR are now the standard first-line treatment for these patients, as these agents provide significant survival benefits by shutting down oncogenic EGFR signals. Afatinib (AF), an irreversible inhibitor of the WT and mutant EGFR, ErbB2, and ErbB4, is one of the second-generation EGFR-TKIs. It prevents signals from all potential homo- or heterodimers of ErbB family receptors [4,5,6]. In addition to potent inhibition of mutant EGFR oncogenic activities, afatinib is unique in the class in that it also binds to wild-type EGFR with high affinity. Although multiple studies have shown that afatinib may have advantages against cancers of the wild-type (WT)-EGFR [2,7,8,9], this chemical uniqueness is underexplored for clinical utility.

Although targeted medicines have made a greater impact thus far, immunotherapy has the potential to make a greater, longer-lasting contribution since it offers the chance of a long-term cure for some patients with metastatic disease. Targeted therapies, however, have had a greater impact thus far because, for the subset of lung cancer patients with driver mutations (often nonsmokers), persistent response and survival extension therapy are targeted to EGFR, ALK, RAF, ROS, RET, and HER2. After a single tumor-targeting therapy, the incidence of tumor recurrence or metastasis is still very high. PTT and immunotherapy in combination can greatly reduce tumor metastasis. When combined with checkpoint blockage, the tumor-associated antigens produced in situ following photothermal tumor ablation may exhibit vaccine-like properties that result in potent antitumor immune responses, enabling successful cancer immunotherapy [10,11,12].

The use of stimulus-responsive therapies as an addition to drug-based therapy is expanding. For instance, photothermal therapy (PTT) uses heat that is locally generated by near-infrared (NIR) light to promote cancer cell death [13,14,15,16,17,18,19]. The cornerstone of PTT is the identification or development of a suitable photothermal transforming agent (PTA) that can convert NIR light into cancer cell-eradicating heat. This PTA should be multifunctional and nontoxic. Utilizing a variety of photoresponsive nanomaterials, the biological characteristics of the source cells can perform a wide range of tasks, including prolonged circulation, immunological evasion, and disease-relevant targeting [20,21,22,23,24].

Since the PTT efficiency is closely correlated with the quality and availability of PPy contained in the NC, PPy plays a crucial role in this process. To enhance the application of NCs for treating cancer, iron oxide (IO) was incorporated, which facilitates the Fenton reaction resulting in the overproduction of H_2_O_2_ (50–100 μM). Cancer cell death is specifically brought about by heat-accelerated production of endogenous H_2_O_2_ in cancer cells, which results in the in situ production of extremely toxic hydroxyl radicals (•OH). Numerous categories of impressive PTAs, according to researchers, include polymer nanomaterials such as polypyrrole (PPy), polydopamine (PDA), and polyaniline (PA), as well as carbon-based graphene and two-dimensional transition metal dichalcogenides [25,26,27,28,29,30,31,32]. By converting the adsorbed energy into heat under stimulation by NIR light, photothermal agents can kill nearby tumor cells by causing local hyperthermia. However, the lack of geometric specificity makes normal cells within tumor tissues vulnerable to PTT damage. There is an urgent, unmet need to specifically target cancer cells and spare adjacent normal stromal cells during NIR-PTT by directing thermal conversion via the guidance of cancer-specific molecules [33,34,35,36].

In this study, the decision to use PPy as the PTA ensured that heat from NIR irradiation would be effectively absorbed while also serving as the central component of a nanocomposite (NC). Reactive oxygen species (ROS) such as •OH, singlet oxygen (^1^O_2_), and superoxide anions (O_2_^•−^) are produced during catalytic reactions when NIR irradiation is used as cancer catalytic therapy [37,38,39].

Although similar methods including TKIs and photothermal therapies have been investigated in the past, those methods involved a prolonged series of sequential steps using nonbiocompatible materials. PTA nanosystems should have prolonged blood circulation cycles and sequentially target tumor tissues and cell membranes to specifically disrupt tumor cell membranes. They should also be simple to make and extremely biologically safe when used clinically. This kind of cell membrane-targeting nanosystem has, as far as we are aware, been infrequently documented [28,40].

We developed a PPy–IO–AF composite called PIA NCs, a novel type of NC, as a solution to the aforementioned issues. Combining multiple effective cancer treatments into a solitary, well-coordinated system that augments the effectiveness of each distinct anticancer strategy and incorporates a TKI and a capable ROS component into the system should lead to a sequential synergetic scheme with exponential increases in cancer cytotoxicity. As such coordinated advances in many methodologies are linked and successfully deployed, this represents the originality of this NC. Additional tests confirmed the biosafety of the NCs we created and their ability to target cancer cells via the targeting impact of AF, Fenton reaction-inducing IO, and the effective PTA PPy. The multifunctional NC as formulated was therefore anticipated to be utilized in a variety of ways as potential nanoplatforms that can induce the programmed cell death (PCD) of cancer cells.

## 2. Materials and Methods

### 2.1. Materials

Pyrrole, polyethyleneimine, ferric chloride, polyethylene glycol, dichlorodihydrofluorescein diacetate (DCFDA) (Sigma Aldrich, Taipei City, Taiwan), AF (MedExpress), ferrous chloride, sodium hydroxide, hydrochloric acid, N-hydroxy succinimide (NHS), 1-ethyl-3-(3-dimethylamino propyl) carbodiimide (EDC), fetal bovine serum (FBS), phosphate-buffered saline (PBS), Dulbecco’s minimal essential medium (DMEM), trypan blue, cyanine 5 dye, Hoechst 33342, an antibiotic/antimycotic (Thermo Scientific, Taipei City, Taiwan), Annexin V, propidium iodide (PI; Elabscience, Taipei City, Taiwan), and formaldehyde (Bio Man, New Taipei City, Taiwan) were commercially procured.

### 2.2. Methods

#### 2.2.1. Material Synthesis

Synthesis of PIA NCs: A PEGylation procedure was used to produce PIA NCs. Two hours of continual stirring with 5 M EDC and 5 M NHS was employed to activate mPEG-COOH. The solution was then mixed with 20 mg of an aqueous PI nanoparticle (NP) solution and stirred for 24 h, after which 0.1 μM of AF was added, and the mixture was stirred continuously for another 24 h. The synthesis procedure is depicted, (Figure 1A,B), while the dispersion of the NC is depicted in a detailed synthesis procedure of PPy, IO, and PI in the Appendix A.

#### 2.2.2. In Vitro Methods

Cell culture: BCRC (Taiwan) and ATCC supplied A549 human lung adenocarcinoma cells, (EGFR-positive NSCLC cells) and NIH3T3 mouse embryonic fibroblasts (EGFR-negative fibroblast cells). Cells were cultured in DMEM supplemented with 10% FBS and 5% antibiotic/antimycotic (Thermo Scientific, Taipei city, Taiwan) in a humidified incubator at 37 °C and 5% CO_2_.

#### 2.2.3. Characterization of NCs

An NP tracking analysis (NTA) was used to examine the size distribution of the NCs (Nano Sight NS300 with a camera type of scalable complementary metal-oxide semiconductor (sCMOS), UK). Before being examined by the NTA, samples were diluted. The zeta potential was determined using laser Doppler velocimetry (Malvern Zetasizer Nano Series, UK) with diluted samples. Fourier transform infrared (FTIR) analyses of the NPs and drug solutions were recorded in the spectral range of 400–4000 cm^−1^ (ThermoFisher NicoletiS10 FTIR Spectrometer, Taipei City, Taiwan). TEM was used to examine the structure of the NPs (HT-7700, Hitachi, Atlanta, GA, USA). Each NP solution was drop-casted onto a copper mesh surface and dried in a hot air oven. SEM was used to examine the surface morphology of dried NC particles (Hitachi SU3500). The technique of energy-dispersive (EDS) X-ray analysis was used to quantify NPs using SEM. The NPs were activated and examined using an EDS X-ray spectrophotometer. Ultraviolet (UV)-visible (Vis)-NIR absorption spectra of NPs were measured at ambient temperature (Jasco V-770 spectrophotometer, Taiwan), and samples were diluted before examination.

#### 2.2.4. Photothermal Properties of NCs

By subjecting PIA NCs in an aqueous dispersion to NIR irradiation (2.45 W/cm^2^ at 808 nm) for different periods at concentrations ranging from 0 to 1 mg/mL and monitoring the change in temperature registered, the photothermal absorption capability of the PIA NCs was examined using a thermocouple (Lutron, TM-925, Eclife, Taipei City, Taiwan) and a thermal camera (SEAT MET-FLTG300+2). The increase in temperature at each concentration was then calculated using results from an additional sample that had no NCs and exposed to radiation under identical conditions. The ratio of the gain in the fluid’s internal energy to the total light intensity that was incident on it was used to measure the photothermal conversion efficiency.

The formula of photothermal conversion efficiency is [fluid mass (g) × (fluid heat capacity (4.2 J/(g/°C)) × average temperature increase (°C)]/[power density of NIR (W/cm^2^) × irradiated time (s) × irradiated area (cm^2^)] × 100 [28].

#### 2.2.5. In Vitro Cellular Uptake

Cy5 NIR fluorescent dye was coupled to the NCs for fluorescence detection to determine the accumulating effect of the PIA NCs. The Cy5-NHS ester was briefly added to the NC solution. The NC/Cy5 mixture was magnetically stirred for 24 h at room temperature. Dialysis was used for 24 h to remove the excess Cy5 that was present in the solution (MWCO 3500). A549 and NIH3T3 cells were grown at a concentration of 10^5^. Cells were given a 1-h treatment with NC/Cy5. Cell nuclei were then counterstained with Hoechst 33342 after the cells had been rinsed with PBS. Using a fluorescence microscope with a fluorescent wavelength excitation and emission for Hoechst 33342 and Cy5, respectively, the images of stained cells were captured.

Additionally, to get a more accurate picture of how the PIA NCs were taken up by cells, a TEM analysis was done. Cells were grown at a concentration of 10^5^ cells/dish in a special TEM two-chamber dish. The medium was replaced with a medium containing the NCs and incubated for 24 h. Cells were then washed with the PBS solution, fixed using a fixation buffer, and then washed again with PBS. After storage at 4 °C, cells were observed by TEM.

#### 2.2.6. In Vitro Cytotoxicity

The cell viability of A549 and NIH3T3 cells was assessed using an MTT test following each treatment to assess the efficacy of the synergetic ROS/PTT NC treatments. In brief, 96-well plates with a cell density of 10^5^ cells/well were seeded with A549 and NIH3T3 cells, and they were then incubated for 24 h. Following PBS washing, cells were subjected to several NC treatments at concentrations ranging from 0 to 1.00 mg/mL in DMEM for 24 h. Cells were exposed to NIR irradiation (2.45 W/cm^2^ at 808 nm) for 1 min and without NIR irradiation after NC treatments. Thereafter, an MTT solution (0.25 mg/mL in PBS) was added and incubated for 2 h at 37 °C. Cells were then lysed using 200 L of dimethyl sulfoxide, and a microplate reader was used to measure the solution’s absorption (570 nm). The percentage of untreated cells that had been absorbed was used to evaluate the in vitro cell vitality.

#### 2.2.7. In Vitro ROS Generation by NC

ROS immunofluorescence (IF) labeling was carried out to evaluate the PIA NCs’ potential to produce ROS in cancer cells in the presence and absence of NIR. Cells were seeded in a confocal dish at a density of 10^5^ cells and incubated for 24 h. Cells were washed with PBS before being exposed to NCs in DMEM. Cells were exposed to NIR irradiation (2.45 W/cm^2^ at 808 nm) for 1 min and without NIR irradiation after NC treatments, followed by PBS washing, and DCFH-DA was used to stain ROS. Cells were fixed with paraformaldehyde, and nuclei were counterstained with Hoechst 33342 and observed under a fluorescent microscope (FM). Additionally, the process of proving that IO played a major role in ROS production is detailed in the Appendix A.

#### 2.2.8. Assessment of Apoptosis

A flow cytometric analysis was conducted to detect the ability of the NCs to cause apoptosis. A549 and NIH3T3 cells were cultured at a density of 10^5^ cells, followed by 24 h of incubation. After washing with PBS, cells were incubated with NCs for 24 h. Cells were exposed to NIR irradiation (2.45 W/cm^2^ at 808 nm) for 1 min and without NIR irradiation after NC treatments. After irradiation, cells were incubated for another 1 h. After removing treated cells, cells were centrifuged and then resuspended in PBS and counted. The cell concentration was 10^6^ cells, and the supernatant was discarded after centrifugation at 300× *g* for 5 min. The pellet was then resuspended in Annexin V binding buffer. Cells were stained with Annexin V-FITC and PI, and the mixture was incubated in the dark for 15 min at room temperature. After gentle vortexing, cells were incubated at room temperature for 15–20 min in the dark. Cells were cytometrically flowed and examined within 1 h.

Additionally, Hoechst 33342 and PI IF staining were used to visualize apoptotic cells after NC treatments. A549 cells were cultured at a density of 10^5^ cells, followed by 24 h of incubation. After PBS washing, cells were incubated with NCs for 24 h. Cells were exposed to NIR irradiation (2.45 W/cm^2^ at 808 nm) for 1 min and without NIR irradiation after NC treatments. PI was added to the cells, followed by 5 min of incubation. Nuclei were counterstained with Hoechst 33342 and observed under an FM.

#### 2.2.9. Statistical Analysis

Experimental results are presented as the average values and standard deviations (±SD). To determine whether statistically significant differences existed in the data, a two-way analysis of variance (ANOVA) for multiple-group comparisons was used, with *p* > 0.05 considered nonsignificant (n.s.), and significance indicated by * *p* < 0.05, ** *p* < 0.005, *** *p* < 0.001, and **** *p* < 0.0001. ImageJ software 1.53f51 was used to quantify the images.

## 3. Results

### 3.1. Preparation and Characterization of PIA NC

The NTA of the PIA NCs revealed that the particle size of the PEI-PPy NPs was 90 nm, and that of IO NPs was 78 nm, which agrees with a review in the literature [41,42]. The particle size of PI NPs was 140 nm indicating their conjugation [43,44], and that of PIA NC was 243 nm. When AF has loaded onto the surface, the size increases, which is reasonable. Conjugation of distinct particles was proven by the increase in particle size. PPy is a conductive polymer that is widely used for biomedical applications [43,45]. In the present study, PPy was synthesized by modifying the existing protocol [46]. We were able to create regular-sized NPs using our modified protocol (Figure 1C).

The zeta potentials of PPy NPs and IO NPs were +42 and −40 mV, respectively, and the charge of PI NPs was +50.6 mV, showing that PEI-PPY and IO had been conjugated. PIA had a zeta potential of +35 mV. This proves that AF had been encapsulated around the NPs. The charge of the NPs was evaluated and found to be of superior quality, with PPy positively charged and IO negatively charged. When PEI and PPy, both of which were positively charged, were combined with IO, the result was a positively charged PI. The charge of the NCs was unaffected by the addition of AF (Figure 1D) [47,48].

The FTIR study depicted the establishment of the NCs’ bonds. On the 3295 cm^−1^ peaks, substantial O–H stretching was seen, with significant C=C stretching at 1637 cm^−1^, and modest C–O–H bending at 1444 cm^−1^. C–O stretching, C–O bending, C–H bending, and Fe–O bonds all had strong appearances at 1244, 1016, 632, and 576 cm^−1^, respectively. The characteristic peak for the NPs proved that the components were conjugated. The production of PIA NCs was shown by the presence of characteristic peaks of PPy, IO, and AF in the samples (Figure 1E).

PPy NPs were found to have a spherical morphology, but IO NPs had a tiny packed cuboidal morphology, as shown by TEM. PI NPs were seen to have a round shape, as did PIA with the drug encapsulating the NPs. The surface morphologies of PPy, IO, PI, and PIA were determined by SEM. Spherical and cuboidal structures were visible on PPy and IO, respectively. Both spherical and cuboidal structures were conjugated in PI. The encapsulated PIA NCs appeared as a sheet-like layered drug (AF) coated onto a cuboidal metal oxide (IO) and a spherical conductive polymer (PPy). The conjugation of IO and PPy with AF to form PIA NCs was validated and confirmed using TEM and SEM images, which showed the surface morphology and was conjoined with FTIR, NTA, and UV (Figure 2A,B) [46,49,50,51,52,53,54].

An elemental analysis of PIA was performed. Since chlorine was already conjugated in the NPs and was not entirely rinsed out during dialysis, the element displayed a distinctive high peak for PPy. Na, Fe, and Cl were all present in IO. The electron dispersion spectrum (EDS) corroborated the presence of C, N, O, and Fe as major elements of PIA NCs. Despite this, the EDS revealed added peaks of F and Cl with C, N, O, and Fe, indicating AF sorption onto the PI composite (Figure 2C,D). PPy NPs had peaks between 400 and 900 nm, while the IO NPs peaked at around 390–800 nm, making them active in the NIR area. The presence of PI NPs in the NIR range demonstrated that the PIA NCs had a broad spectrum in this region, making them a good option for a photothermal study. Despite having a significant peak in the UV region, this did not contribute to any of the NC’s photothermal properties. UV-Vis spectral analysis confirmed that the NCs were active in the NIR region of the spectrum making it a potential photothermal representative (Figure 3A). The results of the UV-Vis analysis of PPy, IO, and PI NPs are given in the Appendix A (Appendix A).

### 3.2. Photothermal Studies

An 808 nm NIR laser was used for distinct periods to examine the heating absorption capabilities of the NCs at various concentrations. At a laser power level of 2.45 W/cm^2^ and PIA samples exposed at a concentration of 1 mg/mL, the temperature of NCs surged to 52.7 °C, a threshold for a perfect temperature to induce hyperthermia that typically kills cancer cells. The temperature rise was proportional to the concentration and was time dependent. Based on data obtained after the NC’s NIR laser heating, the photothermal conversion efficiency of the PIA NC solution was estimated to be 29.71%. This method could be successful for treating cancerous cells (Figure 3B,C). Data of UV-Vis irradiation and the PTT evaluation of PPy and PI are provided in the Appendix A [55,56].

### 3.3. In Vitro Cellular Uptake

The ability of a nanosystem to target a particular area when necessary is one of its key properties. Due to electrostatic interactions that occurred between the positively charged NCs and negatively charged cell membranes of cancer cells, it was anticipated that the positively charged NCs would exhibit increasing levels of cellular uptake. Images showed a significant Cy5 fluorescent signal in the group of A549 cells treated with Cy5-PIA NCs. In the group of NIH3T3 cells treated with Cy5-PIA NCs, a notably lower signal was found. When comparing the various cells, it was shown that A549 cells took up more NCs due to WT-EGFR overexpression, whereas NIH3T3 cells took up fewer or no NCs due to the reduced EGFR expression. The TEM analysis showed the same result. A comparison was made with control cells with no treatment (Figure 4A,B).

### 3.4. In Vitro Cytotoxicity

The cytotoxicity of nanosystems is a crucial consideration when evaluating their potential for use in anticancer applications. In the absence of laser irradiation, conventional photothermal agents need to be biocompatible, but after exposure to light irradiation, they should display an increase in cytotoxicity. There is, however, a compelling need to address these issues while ensuring the development of nanomaterials that prioritize safety, biocompatibility, and toxicity given that intracellular ROS generation has proven to be extremely useful as a treatment/facilitator of cancer cell cytotoxicity without having a significant impact on normal cells.

The MTT experiment showed minimal to no cytotoxicity at various NP concentrations (0–1 mg/mL) of PPy, PI, and PIA alone. The laser-induced cell-killing effect of PIA NCs was, however, more substantial compared to the effect without irradiation. The highest efficiency was seen with PIA NCs at a concentration of 1 mg/mL and a laser effect of 2.45 W/cm^2^. Compared to NIH3T3 cells, PIA NCs were more effective with A549 cells. As the concentration of the treated NPs increased, the percentage of cell viability rapidly dropped. Most PTT is extremely favorable since it allows for more effective targeting of a wide range of lung cancers, regardless of target enrichment. A power density of 2.45 W/cm^2^ for 1 min was sufficient to kill cells [57]. Although EGFR is the relevant target of NSCLC therapy, the efficacy of EGFR-targeted therapies has not been proven in either preclinical models or clinical studies [58].

In A549 and NIH3T3 cells, the cytotoxicity of PIA (±NIR)-based therapies was examined using an MTT assay, with a control group consisting of no additional NCs. Even at the highest concentrations, there was no effect on the viability of cells when no heat treatment was used, indicating that the main cytotoxic effect of NCs is the production of ROS by the Fenton reaction when NIR is applied. The cell death rate of A549 cells was 71% compared to 11% for NIH3T3 cells when NIR was applied (Figure 5A). This was a combination of the targeting effect of NCs specifically toward EGFR-expressing cells and ROS generated when NIR was applied. These inhibition rates show how much better NCs worked in concert compared to individual treatments with PTA PPy and PI separately (Appendix A).

### 3.5. ROS Generation by PIA NCs

For cancer cells to survive, oxidative stress levels and their defenses against the buildup of ROS during carcinogenesis are crucial [59]. A549 cells are cancer cells, and NIH3T3 are normal cells. ROS were produced in higher amounts in A549 cells, and NIH3T3 cells also produced them but in comparatively lower amounts when subjected to PIA + NIR treatment. The Fenton reaction can be employed by iron ions from IO in an acidic milieu to increase levels of ROS in cancer cells. The outcome revealed that higher levels of NC + NIR exposure caused cancer cells to produce more ROS (Figure 5B and Appendix A) [28,60]. Furthermore, the Fenton reaction, one of the most basic processes in treatment with PIA NCs, was independently investigated in cancer cells using IO alone to corroborate its role in creating ROS (Appendix A).

### 3.6. Apoptosis Induced by NCs

The effects of NPs can cause aberrant cell death. For cancer cells to survive, oxidative stress levels and the cells’ defenses against the buildup of ROS during carcinogenesis are crucial [59,61,62]. The flow cytometric analysis showed apoptosis rates in cancer cells and NIH3T3 cells when treated with NC ± NIR. The flow cytometric analysis revealed that 54% of A549 cells and 26% of cells had reached the necrotic and apoptotic phases. The number of viable cells was less than a quarter of a percent. Viability levels of cells that had been treated with NIR or PIA were only 50% and 80%, respectively. The NIH3T3 cells were apoptotic, and necrosis was 0.6% and 5.3%, respectively. Only 10% of NIH3T3 cells had undergone necrosis, meaning that 90% were alive. Viability levels were 65% and 81%, respectively, when only NIR and PIA were used. PIA binding was effective against A549 cells but not against NIH3T3 cells. In NSCLC, the use of EGFR TKIs is a clinically recognized treatment option, particularly for tumors with a sensitizing EGFR kinase domain mutation. TKI treatment with a single drug, on the other hand, does not end the receptor’s oncogenic effects on cell proliferation and apoptosis induction (Figure 6B) [63].

Additionally, fluorescent microscopy was used to compare the number of late apoptotic cells in the treatment group to the number of viable cells in the control group and showed a greater number of late apoptotic cells. Hoechst 33342 positivity and PI negativity indicated living cells. Hoechst 33342-positive/PI-positive cells indicated the presence of late apoptotic cells. The majority of cells treated with PIA and NIR alone did not reach apoptosis. Cells that were treated with NCs along with NIR underwent apoptosis [63]. As more dead cell signals were detected with PIA + NIR in A549 cells compared to control cells, the overall results were consistent with those attained through the flow cytometric analysis. Smaller numbers of apoptotic cells were only visible after PIA and NIR treatments (Figure 6A and Appendix A).

## 4. Discussion

The dark, black polymer known as PPy is widely utilized in a variety of industries for its efficient photothermal conversion, complete spectrum absorption, and highly efficient and high-energy performance of light to thermal energy conversion. PPy has made significant contributions to biological applications such as tissue engineering, actuators, and biosensors. However, this material has drawbacks that make it challenging to use in biological contexts, including brittleness, poor water solubility, and postsynthesis processing issues. There have been several suggestions made to address these drawbacks, including surface modification of the PPy during the polymerization step using charged polymers conjugated to the surface of the NPs. Ferric ions (Fe^+3^ and Fe^+2^) can serve as oxidizing agents to pyrrole monomers during chemical oxidative polymerization that produces PPy [43,64,65,66].

Since the polarity of the coating polymer affects PPy dispersion, dispersion is worse with coated polymers without polar groups. The PPy interface is often coated with a dispersion polymeric agent in a nanoformulation process, such as polyvinyl acetate, that was previously exposed to polymerized pyrrole while being mechanically stirred, to resolve the dispersion problem [46]. Pyrrole monomers were self-polymerized into PPy utilizing ferric ions as the oxidant and polyethyleneimine (PEI) as the stabilizer [36]. Through electrostatic interactions between PPy and IO, subsequent conjugation of IO into the NCs greatly improved their targeting capabilities. Additionally, IO conjugated into the NCs gave it anticancer capabilities through ROS production and made it easier to manage the size because alterations in the amount of conjugated IO directly influence the size [67,68]. As a TKI that has been widely used to treat both WT and mutant EGFR, AF helped in targeting WT EGFR-expressing NSCLC [69,70].

In the tumor microenvironment, polymeric NPs’ size was shown to have a significant impact on the rate of intracellular absorption, with higher cellular uptake with smaller particle sizes. It was proposed that various mechanisms play roles in the uptake of polymeric NPs by cells. NPs of around 200 nm are likely to be ingested through receptor-mediated endocytosis [71,72,73,74,75]. The polymer backbone’s C=C, N–H, and C–N bonds are, respectively, represented by peaks at 1096, 1620, and 3401 cm^−1^. C–H and –OH bonds in PEI were evidenced by respective peaks at 2343 and 2923 cm^−1^. Fe–O bonds were represented by peaks between 400 and 700 cm^−1^. Due to the conversion of a secondary amine into a stable imide (C=N) bond, AF exhibits the majority of its characteristic peaks at 1640 cm^−1^ [76].

The average size of the NCs detected by TEM was approximately comparable to that determined by the NTA, suggesting that those NCs were mainly individually dispersed in the aqueous solution. The PEG coating on the NP surface and AF loading were most likely responsible for the higher NTA-observed diameters compared to the TEM-measured diameters. IO NPs showed a compact cuboidal shape, as demonstrated by TEM, in contrast to the spherical morphology of PPy NPs. PIA with medication encasing the NPs and the round form of PI NPs were both visible. SEM was used to demonstrate the surface morphology of PPy, IO, PI, and PIA. On PPy and IO, respective cuboidal and spherical structures were seen. There were coupled cuboidal and spherical structures in PI [43].

The optical characteristics of the UV-Vis-NIR absorbance of PIA NCs revealed the presence of signals linked to IO in the UV band at 300–400 nm, while NIR absorption of PPY and its photothermal characteristics were connected to the vast absorption band at 700–900 nm in the NIR region [36,43]. The temperature of PIA NCs at 1 mg/mL was estimated to be 52.3 °C, which was significantly higher than the temperature required for photothermal therapy (42–46 °C) [77,78]. The in vitro cellular uptake of NCs was higher in EGFR-expressing cancer cells than in normal cells. This further proved that PIA NCs can specifically target A549 cells and, upon NIR irradiation, cells can successfully be ablated via apoptosis. When combined with PTA, the efficiency of AF increased to killing cancer cells instead of merely restricting their growth. On the other hand, specifically targeted cancer cells were effectively killed with limited hyperpyrexia. The ROS-generated Fenton reaction by IO in the NCs effectively contributed to apoptosis. The flow cytometric analysis showed that PIA NCs together with NIR had a higher rate of apoptosis than individual treatments [28,60,79].

Overall, our findings indicated that PIA NCs along with NIR can be an effective method for ablating EGFR-expressing cancer cells with minimal injury to surrounding cells. The lower cytotoxicity and efficient targeting of specific sites and multiple qualities could be a better choice for treating cancers that express specific molecules. Results from the current study suggest that the combination of PTA PPy, Fenton reaction-inducing IO, and the TKI AF, which targets the WT EGFR in combination with NIR, can be a potential cancer treatment.

## 5. Conclusions

Long regarded as a remarkable method for successfully treating cancers, noninvasive photothermal therapy uses nanomaterials to treat cancer. Like any other treatment, it has several enduring challenges that prevent it from being fully effective. These include the fact that it has no regional effects, damages healthy tissue, and does not completely ablate cancer. The developed PIA NCs proved to be novel, easily fabricated NCs that can improve cytotoxicity and target cancer cells while protecting healthy cells. The capacity to accumulate only on cancer cells, leaving out normal surrounding cells and ROS production via the Fenton reaction when irradiated with NIR, led to apoptosis of cancer cells, which was strong support for hyperthermia treatments. These NCs had several capabilities, such as targeting, ROS production, and causing apoptosis only to cancer cells. Thus, PIA NCs are a promising cancer therapeutic idea for particular molecularly expressed cancers.

## Figures and Tables

**Figure 1 cancers-14-05043-f001:**
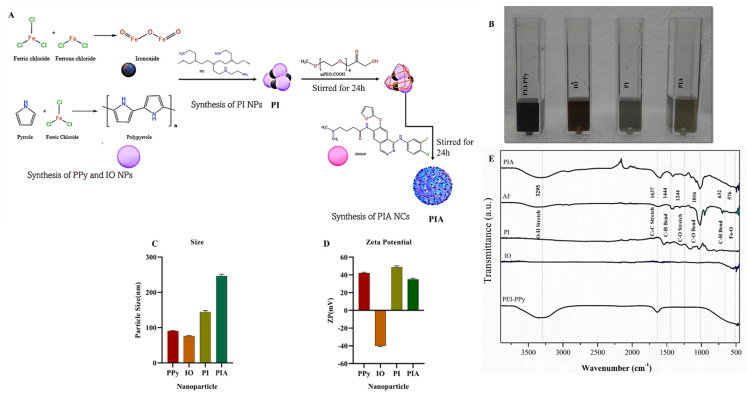
(**A**) Schematic illustration of the synthesis of PIA NC. (**B**) Photograph of dispersed NPs. (**C**) Size distribution of PEI-PPy NPs was 90 nm, IO NPs was 78 nm, PI NPs was 140 nm, and PIA NC was 243 nm. (**D**) The zeta potential of PPy NPs and IO NPs was +42 and −40 mV, respectively, and the charge of PI NPs was +50.6 mV, showing that PEI-PPY and IO had been conjugated. PIA had a zeta potential of +35 mV. (**E**) FTIR analysis of NPs showed on the 3295 cm^−1^ peaks, substantial O–H stretching was seen, with significant C=C stretching at 1637 cm^−1^, and a modest appearance of C–O–H bending at 1444 cm^−1^. C–O stretching, C–O bending, C–H bending, and Fe–O bonds all had strong appearances at 1244, 1016, 632, and 576 cm^−1^, respectively.

**Figure 2 cancers-14-05043-f002:**
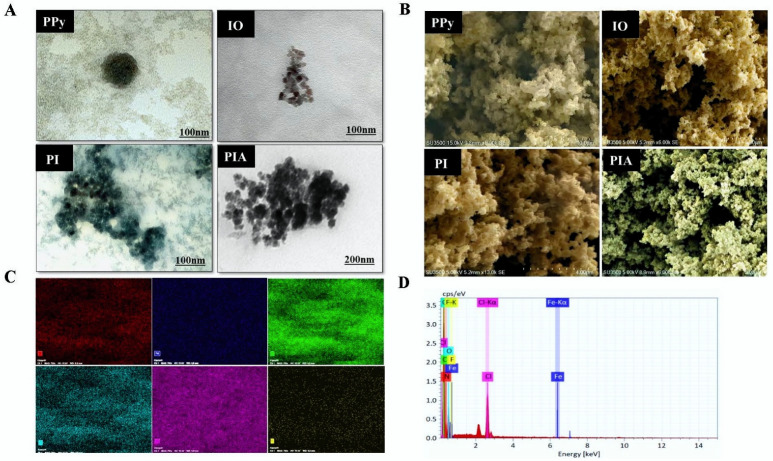
(**A**) TEM image of PPy.IO, PI, and PIA respectively. PPy NPs were found to have a spherical morphology, but IO NPs had a tiny packed cuboidal morphology, as shown by TEM. PI NPs were seen to have a round shape, as did PIA with the drug encapsulating the NPs. (**B**) SEM analysis of PPy, IO, PI, and PIA. Spherical and cuboidal structures were visible on PPy and IO, respectively. Both spherical and cuboidal structures were conjugated in PI. The encapsulated PIA NCs appeared as a sheet-like layered drug (AF) coated onto a cuboidal metal oxide (IO) and a spherical conductive polymer (PPy). (**C**,**D**) EDS analysis of PIA NCs in the presence of Na, Fe, Cl, and added peaks of F and Cl with C, N, O, and Fe, indicating AF sorption onto the PI composite.

**Figure 3 cancers-14-05043-f003:**
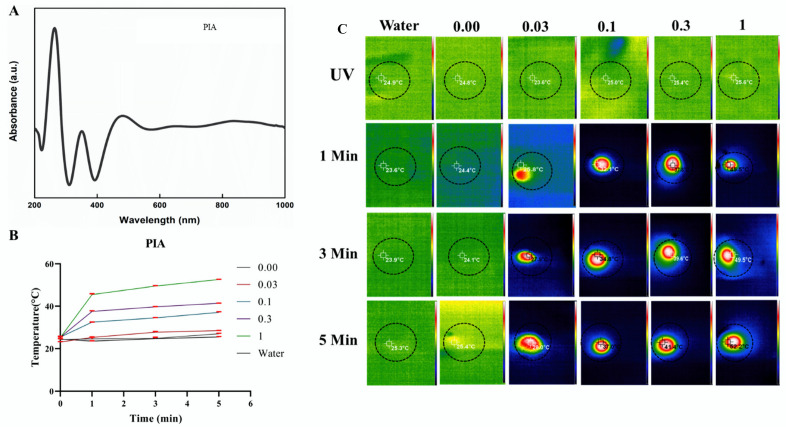
(**A**) UV-VIS-NIR spectra of PIA NC were active in the NIR region of the spectrum, making it a potential photothermal representative. (**B**) Photothermal graph of PIA NC at various times and concentrations (the red line indicates error bars). (**C**) The thermal camera images of PIA NC. At a laser power level of 2.45 W/cm^2^ and PIA samples exposed at a concentration of 1 mg/mL, the temperature of NCs surged to 52.7 °C *(n* = 3).

**Figure 4 cancers-14-05043-f004:**
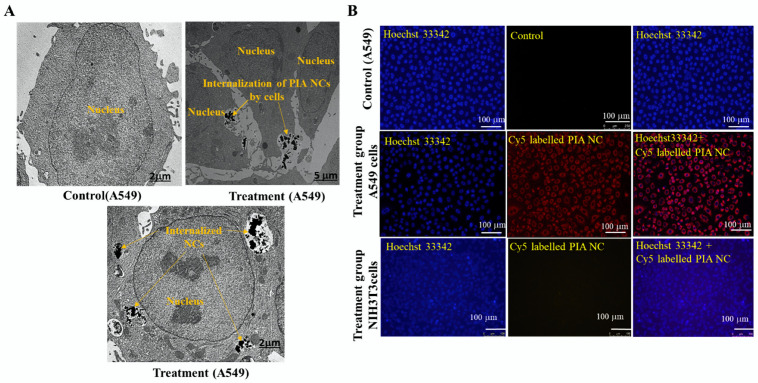
(**A**) The TEM image of control cells, NC being internalized by A549 cells and internalized NC by the A549 cells. (**B**) The FM images of control cells, Cy-5-labeled PIA NC by A549 cells, and Cy-5-labeled PIA NC by NIH3T3 cells (*n* = 3). Images showed a significant Cy5 fluorescent signal in the group of A549 cells treated with Cy5-PIA NCs. In the group of NIH3T3 cells treated with Cy5-PIA NCs, a notably lower signal was found.

**Figure 5 cancers-14-05043-f005:**
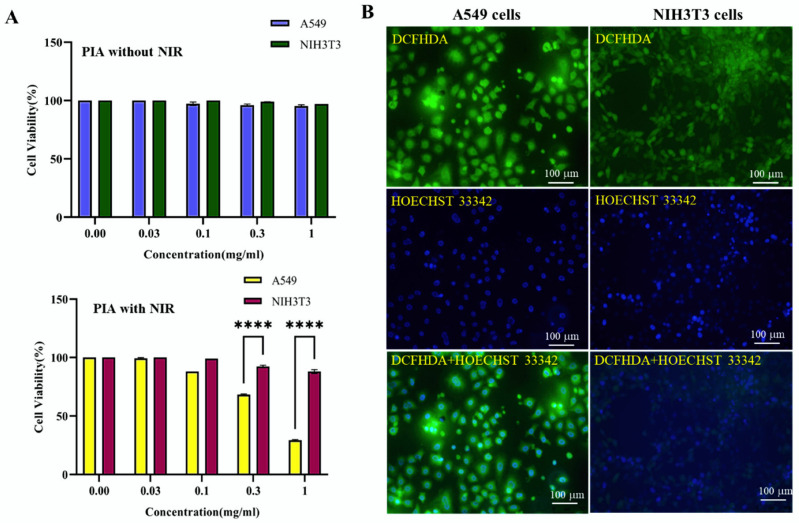
(**A**) Cytotoxicity data of PIA NC with and without NIR irradiation. Even at the highest concentration, there was no effect on the viability of cells when no heat treatment was used, indicating that the main cytotoxic effect of NCs is the production of ROS by the Fenton reaction when NIR is applied. The cell death rate of A549 cells was 71% compared to 11% for NIH3T3 cells when NIR was applied. (**B**) The ROS production by A549 cells and NIH3T3 cells after PIA + NIR treatment *(n* = 3, **** *p* < 0.0001).

**Figure 6 cancers-14-05043-f006:**
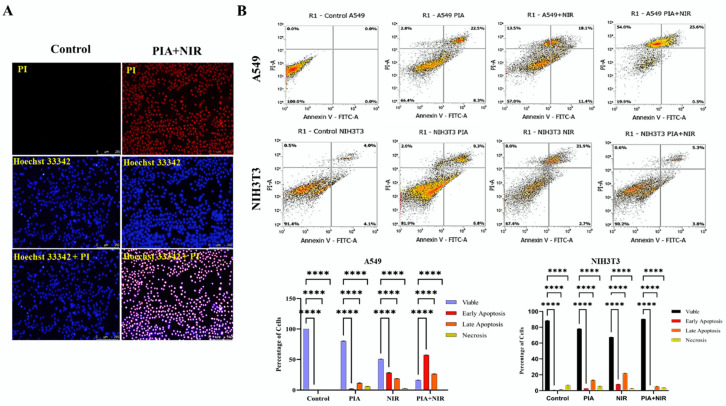
(**A**) The FM analysis of the apoptosis in A549 cells in control and under PIA + NIR treatment. (**B**) Flow cytometry data of A549 cells and NIH3T3 cells under different treatment conditions and graphical representation of the PCD under different treatments; 54% of A549 cells and 26% of cells had reached the necrotic and apoptotic phases. Viability levels of cells that had been treated with NIR or PIA were only 50% and 80%, respectively. The NIH3T3 cells reached apoptotic, and necrosis were 0.6% and 5.3% respectively. Only 10% of NIH3T3 cells had undergone necrosis, meaning that 90% were alive. Viability levels were 65% and 81%, respectively, when only NIR and PIA were used. PIA binding was effective against A549 cells but not against NIH3T3 cells *(n* = 3, **** *p <* 0.0001).

## Data Availability

The data presented in this study are available in this article.

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
