# Peer review of "Molecularly Targeted Photothermal Ablation of Epidermal Growth Factor Receptor-Expressing Cancer Cells with a Polypyrrole–Iron Oxide–Afatinib Nanocomposite"

_cancers, 2022, doi:10.3390/cancers14205043_

Round 1
Reviewer 1 Report
The research article entitled “Molecularly Targeted Photothermal Ablation of Epidermal Growth Factor Receptor-Expressing Cancer Cells with a Polypyrrole-Iron Oxide-Afatinib Nanocomposite” investigated the photothermal ablation capability of Polypyrrole-Iron Oxide-Afatinib Nanocomposite in the invasive treatment of cancer. The article has many grammatical and sentence errors, and the language organization needs to be improved. For these reasons, I conclude that the paper should undergo minor revision.
1. Abstract needs to be improved by providing more quantitative data
2. The introduction is good but very general in nature. More elaborately.
3. Authors may provide greater details of other cancer therapies with their merits and demerits. How immunotherapy is photothermal therapy better than others
4. Please provide a higher resolution of Figures 5 and 7
5. Typographical errors can be avoided. The language and grammar used throughout the manuscript need to be improved. Specific attention needs to be given to this which will improve the standard of the manuscript.
Author Response
Respected Reviewer,
Please see the attachment below.

Reviewer 2 Report
The authors have submitted an interesting article entitled "Molecularly Targeted Photothermal Ablation of Epidermal Growth Factor Receptor-Expressing Cancer Cells with a Polypyrrole-Iron Oxide-Afatinib Nanocomposite" which deals with the synthesis of a new photothermal polymeric-based nanocomposite for cancer treatment. I suggest this article be published after a major revision.
Comments:
1- First of all, I would like to recommend authors to design a better “Graphical Abstract” for this study to better show the whole story in a simple and informative manner. To this end, you can easily use the Biorender website to make a simple, but intuitive and informative illustration. The mechanism of cancer cell ablation and a few images of the final cell study is suggested to be included in your graphical abstract.
2- Please carefully revise the manuscript to remove typos, grammatical errors, and vague sentences. Some of the sentences are unnecessarily long (like the very first sentence of the introduction) which makes it difficult and boring for the readers to follow them.
“For example Figure 2, E … should change to Transmittance (a.u.)” and many others.
Please double-check the whole manuscript and revise all.
3- Some figures are of low quality and difficult to read (for example figure 1 (A and E)). Please replace them with high-quality figures.
4- The introduction part has provided a comprehensive and sufficient background of the existing challenge and the current treatment. However, it seems that the last sections and the novelty section are messed up! From line 110, please swap the paragraph (line 121-127) with its previous one (line 110-120) in the order and keep the integrity of the novelty statement.
5- Some of the references in the introduction part or other parts of this manuscript are too old (e.g., 2001, 2003, 2005, etc.) and it is not acceptable at all. A myriad of research bodies has been published in recent years and you can find similar concepts and cite them in your paper rather than more than 2 decades old references. Moreover, in the introduction part and related to photoresponsive materials and the fundamentals of this field, please read and add valuable information from the following paper as well: https://doi.org/10.3390/ijms23042223, https://doi.org/10.1016/B978-0-12-818806-4.00010-3
6- To increase the validity of obtained data, the authors require to repeat all experimental sections (like photothermal activity Figure 4. B and so forth) with at least three replications, do statistical analysis of data, and then provide graphs with error bars.
Author Response
Respected Reviewer,
Please see the attachment below

Round 2
Reviewer 2 Report
The manuscript is well-amended and it is ready for publication. I have no further comments.